# Nutritional Adequacy and Diet Quality Are Associated with Standardized Height-for-Age among U.S. Children

**DOI:** 10.3390/nu13051689

**Published:** 2021-05-16

**Authors:** Kijoon Kim, Melissa M. Melough, Dongwoo Kim, Junichi R. Sakaki, Joonsuk Lee, Kyungju Choi, Ock K. Chun

**Affiliations:** 1Department of Nutritional Sciences, University of Connecticut, Storrs, CT 06269, USA; drkijoon@gmail.com (K.K.); melissa.melough@uconn.edu (M.M.M.); junichi.sakaki@uconn.edu (J.R.S.); 2Department of Food and Nutrition, Sookmyung Women’s University, Seoul 04310, Korea; 3Department of Human Ecology, College of Natural Science, Korea National Open University, Seoul 03087, Korea; kimdow@knou.ac.kr; 4BOM Institute of Nutrition and Natural Medicine, Seoul 05554, Korea; joonsuk.lee@nutrean.com (J.L.); celli96@gmail.com (K.C.)

**Keywords:** height-for-age, children, nutritional adequacy, NHANES, diet

## Abstract

Nutritional status affects linear growth and development. However, studies on the associations between nutritional status, diet quality, and age-standardized height in children are limited. The aim of this study was to assess the relationship between macro- and micronutrient intake and food consumption and height-for-age Z score (HAZ) among US children in the National Health and Nutrition Examination Survey (NHANES). This cross-sectional population-based study included 6116 US children aged 2–18 years. The usual dietary intake of nutrients and food groups was estimated by the multiple source method (MSM) using two-day food consumption data from NHANES 2007–2014. After adjusting for covariates, HAZ was positively associated with intakes of energy, protein, carbohydrate, fat, vitamins A, D, E, B6, and B12, thiamin, riboflavin, niacin, calcium, and iron. Children in the highest tertile of HAZ were less likely to consume lower than the EAR for vitamin E and calcium. Major foods consumed by children with lower HAZ were soft drinks, high-fat milk products, cakes, cookies, pastries, and pies, whereas children with higher HAZ tended to consume low-fat milk products, tea, and low-calorie fruit juice. These findings suggest that adequate nutritional intake, diet quality, and nutrient-dense food are important factors for height in children.

## 1. Introduction

Height is a good overall indicator of a child’s growth and well-being, and stunting in childhood is a critical impediment to human development. Stunting is one of six global nutrition targets that the World Health Organization (WHO) have endorsed for improving maternal, infant and young child nutrition by 2025 [1]. Approximately 162 million children under five years old have been affected by growth faltering globally, which has negative impacts on individuals and societies, including impaired cognitive development [2,3], poor health and reduced productive capacity [4], and increased risk of disease and mortality [5,6].

Growth in childhood may be determined by the cumulative effects of short-and long-term factors, including household and family environment [7,8], infection [9], and nutritional intake [10]. Among several determinants of growth, it is well known that nutritional status in childhood directly affects growth and development [11]. Although extreme hunger is likely to be linked to severe malnutrition, well-nourished populations may not necessarily achieve nutritional adequacy. Adequate intake of energy, protein, and other nutrients plays an important role in growth. Energy and protein restriction have been shown to result in significant decreases in insulin-like growth factor 1 (IGF-1), a key hormone in the promotion of childhood growth, in animal [12] and human [13] studies. Selected studies have reported on the relationships between supplementation of particular nutrients or foods and growth [14,15], yet findings from previous studies on the effects of nutrients, foods, and food groups on skeletal growth are still limited. A randomized controlled trial [14] showed that high-dose vitamin A supplementation improved linear growth among Indonesian preschool children with low serum retinol. However, there are few studies exploring the relationship between dietary intake and linear growth based on the estimation of usual dietary intake. Therefore, the objective of this study was to assess the relationship between height-for-age Z score (HAZ) and usual intake of macronutrients, micronutrients, and food groups among US children, using data from the National Health and Nutrition Examination Survey (NHANES). To our knowledge, this is the first report describing the associations between nutritional status, diet quality, and age-standardized height in a large, representative sample of US children.

## 2. Materials and Methods 

### 2.1. Study Population

This cross-sectional study included 6116 US children aged 2–18 years from NHANES 2007–2014 [16,17,18]. We excluded those with dietary recalls coded as unreliable or incomplete (*n* = 19), those who reported that 24-h diet was unusual, such as “much less than usual” or “much more than usual” (*n* = 3770), those who were on any kind of diet to lose weight or for other health-related reasons (*n* = 209), and those with missing height data (*n* = 125). NHANES study protocols were approved by the National Center for Health Statistics research ethics review board. (Continuation of Protocol #2005-06 for NHANES 2007-2010, Protocol #2011-17 for NHANES 2011-2012, and Continuation of Protocol #2011-17 for NHANES 2013-2014).

### 2.2. Estimation of Usual Dietary Intake

Dietary data were collected from participants through two 24-h dietary recalls. The first 24-h dietary recall was collected during an in-person interview and the second was collected by phone. The usual dietary intake of macronutrients, micronutrients and food groups was estimated by the Multiple Source Method (MSM) using two 24-h dietary recalls from NHANES 2007-2014. MSM is a new statistical method for estimating usual dietary intake on the basis of two or more short-term measurements, such as 24-h dietary recall data. The statistical algorithms of MSM account for intra-individual variation of intake. This method is characterized by a two-part shrinkage technique applied to residuals of two regression models: One for the positive daily intake data and one for the probability of consumption [19]. For the estimation of usual intake using MSM, different models were used for nutrients and food groups. Since the probability of nutrients consumption on a certain day is close to 100%, the probability of consumption was not added as a variable, and all subjects were considered as habitual consumers in the usual intake estimation model. However, as there is a possibility that there is a true non-consumer for food groups, the probability of consumption of the food group was added as a covariate to the model. The most accurate way to assess food group intake is to obtain frequency information for specific food groups through a long-term measurement such as a food frequency questionnaire (FFQ). However, as there is no FFQ in NHANES 2007–2014, the probability of consumption for each food group was calculated from the FFQ data collected in NHANES 2005–2006 and used in the process of estimating usual intake of NHANES 2007–2014 data, alternatively [20]. Participants were also classified as having met or having failed to meet the estimated average requirements (EAR) [21] for nutrients based on the usual intake calculated by MSM from the two days of dietary recalls. Whether participants had met or failed to meet the EAR for nutrients was determined according to age and gender.

Although food groups are chosen based on the first two digits of the USDA 8-digit food code, some foods were disaggregated to identify frequently consumed foods. Milk and milk products were disaggregated into high fat dairy products, low fat dairy products and yogurt. Frequently consumed fruit and 100% fruit juice, such as apple, orange, berry, banana, grapefruit, 100% apple juice, and 100% orange juice were separated from fruits and fruit juice. Nonalcoholic beverages were separated into coffee, tea, regular fruit juice drink, low-calorie fruit juice drink, and total carbonated soft drink.

### 2.3. Estimation of Height-for-Age Z Score

Physical examination data of participants including heights and weights were collected by trained health technicians in the Mobile Examination Center and measured as described in the NHANES Anthropometry Procedures Manual [22]. HAZ was calculated using World Health Organization (WHO) 2006 growth standards [23]. The Z score was estimated by using the L (power transformation for skewness), M (median), and S (dispersion) parameters in Cole’s LMS method [24].

### 2.4. Statistical Analysis

Statistical analyses were performed using SAS software, version 9.4 (SAS Institute Inc., Cary, NC, USA), using SAS survey procedures and the appropriate weight, strata, and cluster variables to account for the complex survey design. Participants were grouped into tertiles based on HAZ. The mean and frequencies according to tertiles of HAZ were calculated across sociodemographic factors. P-values for differences among subgroups were obtained by chi-square test and ANOVA. Odds ratios (ORs) for nutrient intake below the EAR and the corresponding 95% confidence intervals (CIs) were estimated using multiple logistic regression modeling. P-trends were calculated in a model adjusted for age, gender, ethnicity and birth weight. Birth weight and sleep hours were measured using questions of ‘how much sleep do you get (hours)?’ and ‘how much did you weigh at birth?’ Participants were grouped by poverty income ratio (PIR) as follows: <1.3, 1.3≤ to <1.85 and ≥1.85. PIR was defined as ratio of family income to federal poverty level. Physical activity was presented as metabolic equivalent of tasks (MET), which were based on weekly minutes of walking/bicycling and moderate and vigorous recreational activities, and multiplying the number of days per week by the average minutes of activities on a typical day [25]. MET-min/week were determined by multiplying weekly minutes of activities by the assigned MET values. We used the MET values provided by the NHANES manual, where suggested MET values of moderate recreational activities such as walking/bicycling were 4 MET and vigorous recreational activities were 8 MET. Participants who reported no walking/bicycling and no moderate and vigorous recreational activities for at least 10 min continuously were defined as inactive. Physical activity was then categorized as inactive, <500 MET-min/week, and ≥500 MET-min/week. All P-values reported are two sided (α = 0.05).

## 3. Results

The majority of children were white (59.7%), had a PIR >1.85 (56.4%), engaged in at least a modest amount of physical activity (77.4%), were in the healthy weight range (66.1%), and did not use supplements (74.3%) (Table 1). Additionally, the subjects with higher HAZ were more likely to have a higher income, higher birth weight, and higher BMI. Higher HAZ was associated with greater consumption of vitamins A, D, and E, thiamin, riboflavin, niacin, vitamin B6, vitamin B12, calcium, and iron after adjusting for gender, age, ethnicity, and birth weight (Table 2).

HAZ was positively associated with total energy intake but not with percent of energy from each macronutrient. The latter is unsurprising given the close similarities in macronutrient compositions of the diets. Additionally, those in the highest HAZ tertile had greater consumption of protein, carbohydrate, and fat compared to those in the lowest tertile (Figure 1). Among the five major food groups, the consumption of grains and dairy but not fruits, vegetables or meat/protein was greatest for children in the highest HAZ tertile (Figure 2).

Compared to children in the lowest tertile of HAZ, children in the highest tertile of HAZ were 25% less likely to consume below the EAR of vitamin E (OR: 0.75; 95% CI, 0.59–0.95) after adjusting for gender, age, ethnicity and birth weight. With regards to calcium, children in the middle HAZ tertile were 21% less likely to consume below the EAR (OR: 0.79; 95% CI: 0.64–0.97) while children in the highest tertile were 31% less likely to consume below the EAR (OR: 0.69; 95% CI, 0.57–0.85) compared to children in the lowest tertile (Table 3). While the food most consumed (by weight) by children in the highest tertile of HAZ was low-fat milk products, the food most consumed by those in the lowest tertile of HAZ was carbonated soft drinks. Additionally, while those in the lowest tertile of HAZ consumed greater amounts of high-fat milk, cakes, cookies, pastries, and pies, those in the highest tertile of HAZ consumed greater amounts of tea and low-calorie fruit juice drinks (Table 4).

## 4. Discussion

In this nationally representative study, US children aged 2–18 years old with higher HAZ tended to have a higher income and BMI percentile. This result is consistent with previous reports that HAZ was positively associated with household income [26,27,28] and BMI [29]. Although higher BMI was associated with greater HAZ in this study, without follow-up data we could not determine whether children with lower BMIs may eventually attain similar heights as their higher BMI peers. The current study also found that those with higher HAZ had higher birth weight, which is consistent with previous reports showing that HAZ was positively associated with birth weight in the Alive and Thrive Study [30] and in the Brazilian follow-up study [28].

This study found that HAZ is positively associated with energy intake after adjusting for age, gender, ethnicity, and birth weight. This finding is consistent with a previous study reporting that stunted children had a lower energy intake [31]. Several studies showed that lower energy intake was associated with lower levels of bone mineral content (BMC) [32] and IGF-1 [13]. We found that there was a positive association between HAZ and intake of carbohydrate, fats, and protein, which is consistent with previous studies showing that there was a significant positive association between linear growth and carbohydrate [33], fat [34], and protein [31,33]. The current study also found that HAZ was not associated with percentage of energy from any macronutrient. This suggests that total macronutrient and energy intake, rather than the distribution of energy across macronutrients, is associated with HAZ.

This study found that HAZ was positively associated with consumption of vitamin A, vitamin D, vitamin E, thiamin, riboflavin, niacin, vitamin B6, vitamin B12, calcium, iron, and food groups of grains and dairy products after adjusting for age, gender, ethnicity and birth weight. It is important to consider that this finding may, at least in part, relate to the fact that taller children are likely to consume more food than shorter children, which ultimately contributes more to micronutrient intakes. We also found that children in the highest tertile of HAZ were less likely to consume below the EAR of vitamin E and calcium after adjusting for potential confounders. These results are consistent with previous studies reporting that there was a positive association between linear growth and intake levels of vitamin A [11,14,35], vitamin D [32,36], vitamin E [37,38], riboflavin [11,36], vitamin B6, calcium [11,32,36] and dairy products [39]. Several studies [15,40,41] reported that dairy product supplementation stimulates linear growth in both well-nourished and malnourished children, presumably through stimulation of circulating IGF-1 [42,43,44] and enhanced BMC [44,45].

This study found that major foods consumed by children in the lowest tertile of HAZ were soft drinks, cakes, cookies, pastries and pies, and the major foods consumed by those in the highest tertile of HAZ were low-fat milk products, tea, and low-calorie fruit juice drinks. This result showed that children with higher HAZ may have healthier food choices and diet quality compared to children with lower HAZ. Soft drinks, which were popular choices among children with relatively lower HAZ in this study, have previously been associated with lower intake of milk [46,47,48], calcium [48,49], and vitamin A [49], and with lower diet quality [49,50,51], resulting in an increased risk of several medical problems such as asthma [52], hypocalcemia [53,54], and type 2 diabetes [55]. Conversely, greater consumption of low-calorie beverages/tea, which was an important feature of the diets of high HAZ children in this study, has previously been associated with higher diet quality compared to greater consumption of sugar-sweetened beverages [56].

This study is strengthened by its use of a large, nationally representative sample of US children, allowing for the detailed analysis of the relationship between dietary intake and HAZ across participants with varying sociodemographic characteristics. Additionally, we estimated usual dietary intake of nutrients and food groups using MSM, enabling more accurate estimation of usual intake by reducing intra-individual variation. However, this study has some limitations. First, no individual’s growth trajectory can be captured from this study because the present study uses a cross-sectional design. Second, as heights of participants’ parents were not available in NHANES, birth weight as a genetic factor was controlled in our models. Third, because the estimation of usual intake of dietary supplement by the MSM method was not available, dietary supplement was not considered in this analysis. Finally, even though attempts were made to adjust for all relevant covariates available in NHANES, residual confounding factors may still be present.

## 5. Conclusions

In conclusion, a higher HAZ was associated with a greater intake of energy, dairy products, grains, low-fat milk, tea, and micronutrients among US children, suggesting that adequate nutritional intake, diet quality, and nutritious food choices could be beneficial to linear growth in childhood. Future studies should prospectively assess the relationship between dietary intake/nutritional adequacy and height growth in order to better elucidate the relationship’s directionality.

## Figures and Tables

**Figure 1 nutrients-13-01689-f001:**
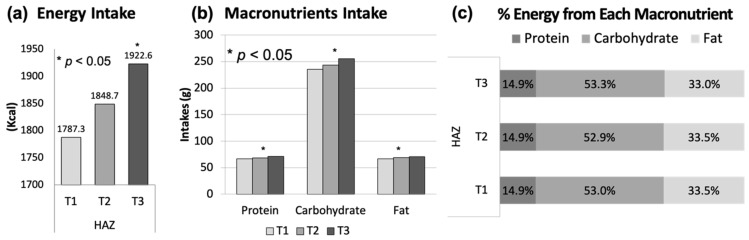
(**a**) Energy intake, (**b**) protein, carbohydrate, fat intake, (**c**) % energy from each macronutrient by tertile of HAZ among US children aged 2–18 years, NHANES 2007–2014 (*n* = 6116) Each tertile was defined as follows: T1 (low HAZ), T2 (medium HAZ), T3 (high HAZ) * Tested by ANOVA and adjusted for age, gender, ethnicity and birth weight.

**Figure 2 nutrients-13-01689-f002:**
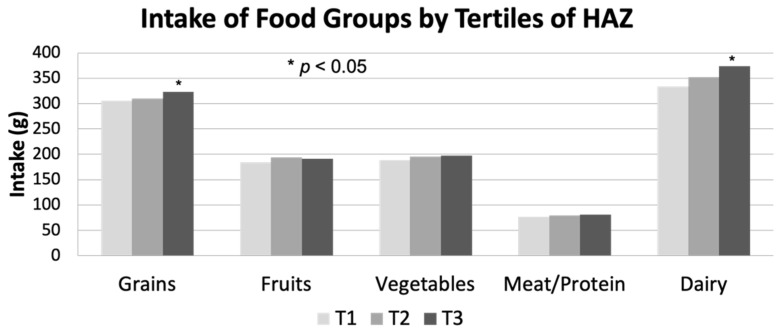
Intake of five food groups by tertile of HAZ among US children aged 2–18 years, NHANES 2007–2014 (*n* = 6116). Each tertile was defined as follows: T1 (low HAZ), T2 (medium HAZ), T3 (high HAZ) * Tested by ANOVA and adjusted for age, gender, ethnicity and birth weight. HAZ, height-for-age Z score.

**Table 1 nutrients-13-01689-t001:** Sociodemographic and lifestyle characteristics by tertile of HAZ among US children aged 2–18 years in NHANES 2007–2014 (*n* = 6116).

			HAZ (Min, Max)						
	Total		T1 (*n* = 2038) (4.57, −0.27)		T2 (*n* = 2039) (−0.27, 0.57)		T3 (*n* = 2039) (0.57, 3.98)		*p*-Value *
	*n*	%	*n*	%	*n*	%	*n*	%	
Gender									0.203
Boys	3125	50.9	1024	50.2	1034	49.1	1067	53.2	
Girls	2991	49.1	1014	49.8	1005	50.9	972	46.8	
Age (y)									0.278
2–6	2281	33.4	721	34.5	788	33.2	772	32.6	
7–12	2210	35.6	678	33.5	726	34.5	806	38.8	
13–18	1625	30.9	639	32.0	525	32.3	461	28.6	
Ethnicity									<0.0001
White	1987	59.7	607	55.5	681	59.8	699	63.5	
Black	1373	12.5	367	11.3	432	11.4	574	14.6	
Mexican-American	1396	13.8	549	17.0	488	14.5	359	10.1	
Other	1360	14.0	515	16.2	438	14.3	407	11.8	
PIR									<0.001
<1.3	2520	33.1	926	40.2	843	31.6	751	27.9	
1.3–1.85	697	10.5	229	10.4	223	10.7	245	10.4	
>1.85	2488	56.4	745	49.4	827	57.7	916	61.7	
Mother’s age when born									0.131
<30	4203	66.3	1476	69.3	1440	68.9	1351	64.1	
30–39	1716	30.8	518	28.4	560	29.6	638	34.0	
≥40	197	2.9	44	2.3	39	1.5	50	1.9	
Physical activity									0.927
Inactive ^a^	245	12.0	104	13.4	78	11.2	63	11.4	
<500 MET-min/wk	213	10.6	89	10.5	66	10.3	58	10.9	
≥500 MET-min/wk	1398	77.4	514	76.1	448	78.5	436	77.7	
BMI (percentile)									<0.0001
<5	209	3.4	99	5.1	69	3.2	41	1.9	
5–84.9	3997	66.1	1496	74.6	1352	66.8	1149	57.4	
85–94.9	910	15.0	259	10.7	303	15.9	348	18.1	
≥95	1000	15.5	184	9.6	315	14.1	501	22.5	
Supplement use									0.339
Yes	1286	25.7	448	27.7	429	24.1	409	25.6	
No	4825	74.3	1588	72.3	1608	75.9	1629	74.4	
Sleep hours (n, hours)	743	7.6	331	7.4	226	7.6	173	7.7	0.073
Birth weight (n, kg)	5373	3.1	1661	3.0	1769	3.1	1832	3.3	<0.0001

^a^ Not engaging in any walking/bicycling or moderate or vigorous recreational activities for at least 10 min continuously in a typical week. HAZ, height-for-age Z score; MET, metabolic equivalent of tasks; PIR, poverty income ratio; Each tertile was defined as follows: T1 (low HAZ), T2 (medium HAZ), T3 (high HAZ) * Tested by ANOVA or chi-square test.

**Table 2 nutrients-13-01689-t002:** Daily average micronutrient intakes by tertile of HAZ among US children aged 2–18 years in NHANES 2007–2014 (*n* = 6116).

	HAZ (Min, Max)			*p* for Trend *
	T1 (*n* = 2038) (−4.57, −0.27)	T2 (*n* = 2039) (−0.27, 0.57)	T3 (*n* = 2039) (0.57, 3.98)	
	Mean (SE)	Mean (SE)	Mean (SE)	
Vitamin A (μg/d)	585.8 (7.2)	611.6 (10.5)	646.3 (9.2)	<0.01
Vitamin C (mg)	78.7 (1.4)	82.0 (1.8)	82.2 (1.4)	0.103
Vitamin D (μg/d)	5.7 (0.1)	5.9 (0.1)	6.2 (0.1)	<0.05
Vitamin E (μg/d)	6.2 (0.1)	6.5 (0.1)	6.7 (0.1)	<0.05
Thiamin (mg/d)	1.5 (0.0)	1.5 (0.0)	1.6 (0.0)	<0.001
Riboflavin (mg/d)	1.9 (0.0)	2.0 (0.0)	2.1 (0.0)	<0.001
Niacin (mg/d)	20.6 (0.2)	21.2 (0.3)	22.2 (0.4)	<0.05
Vit B6 (mg/d)	1.7 (0.0)	1.8 (0.0)	1.8 (0.0)	<0.01
Vit B12 (mg/d)	4.8 (0.1)	5.0 (0.1)	5.2 (0.1)	<0.05
Folate (μg/d)	357.1 (4.4)	366.0 (5.9)	383.5 (5.2)	0.099
Calcium (mg/d)	986.6 (10.0)	1017.7 (13.1)	1085.1 (13.6)	<0.0001
Iron (mg/d)	13.7 (0.2)	14.1 (0.2)	14.6 (0.2)	<0.05
Zinc (mg/d)	9.9 (0.1)	10.2 (0.1)	10.6 (0.1)	0.103

HAZ, height-for-age Z score. Each tertile was defined as follows: T1 (low HAZ), T2 (medium HAZ), T3 (high HAZ) * *p* for trends were calculated in a model adjusted for age, gender, ethnicity and birth weight.

**Table 3 nutrients-13-01689-t003:** Odds ratios and 95% CIs for consuming below the EAR by tertile of HAZ among US children aged 2-18 years in NHANES 2007-2014 (*n* = 6116).

	HAZ (Min, Max)			*p* for Trend *
	T1 (*n* = 2038) (−4.57, −0.27)	T2 (*n* = 2039) (−0.27, 0.57)	T3 (*n* = 2039) (0.57, 3.98)	
Vitamin A (μg/d)	1.0 (ref)	0.81 (0.56–1.17)	0.74 (0.50–1.09)	0.288
Vitamin C (mg)	1.0 (ref)	0.73 (0.49–1.08)	0.79 (0.52–1.18)	0.249
Vitamin D (μg/d)	1.0 (ref)	0.80 (0.49–1.30)	0.71 (0.45–1.11)	0.296
Vitamin E (μg/d)	1.0 (ref)	0.81 (0.63–1.03)	0.75 (0.59–0.95)	<0.05
Thiamin (mg/d)	1.0 (ref)	1.29 (0.47–3.55)	1.45 (0.57–3.69)	0.702
Riboflavin (mg/d)	1.0 (ref)	1.02 (0.30–3.48)	0.81 (0.18–3.60)	0.949
Niacin (mg/d)	1.0 (ref)	0.19 (0.04–1.03)	0.36 (0.09–1.40)	0.099
Vit B6 (mg/d)	1.0 (ref)	0.53 (0.15–1.92)	1.63 (0.45–5.91)	0.257
Vit B12 (mg/d)	1.0 (ref)	0.34 (0.08–1.46)	1.03 (0.19–5.61)	0.302
Folate (μg/d)	1.0 (ref)	0.96 (0.62–1.48)	1.01 (0.66-1.54)	0.977
Calcium (mg/d)	1.0 (ref)	0.79 (0.64–0.97)	0.69 (0.57–0.85)	<0.01
Iron (mg/d)	1.0 (ref)	1.06 (0.36–3.11)	0.80 (0.27–2.39)	0.886
Zinc (mg/d)	1.0 (ref)	1.31 (0.71–2.43)	0.90 (0.50–1.60)	0.496

CI, confidence interval; EAR, Estimated Average Requirements; HAZ, height-for-age Z score. Each tertile was defined as follows: T1 (low HAZ), T2 (medium HAZ), T3 (high HAZ) * *p* for trends were calculated in a model adjusted for age, gender, ethnicity and birth weight.

**Table 4 nutrients-13-01689-t004:** Comparison of major foods consumed by children aged 2–18 years in the highest HAZ vs. lowest HAZ in NHANES 2007-2014 (*n* = 6116).

	Lowest HAZ (T1)		Highest HAZ (T3)	
Rank	Food Group	Intake (g/d)	Food Group	Intake (g/d)
1	Total carbonated soft drinks	162.6	Low-fat milk products	182.2
2	Mixtures mainly grain	150.4	Total carbonated soft drinks	162.7
3	Low-fat milk products	145.1	Mixtures mainly grain	156.3
4	Regular fruit juice drinks	78.4	Regular fruit juice drinks	83.6
5	Mixtures mainly meat, poultry, fish	74.1	Mixtures mainly meat, poultry, fish	78.5
6	High-fat milk products	56.8	Tea	69.2
7	Tea	51.7	High-fat milk products	43.9
8	100% Orange juice	37.6	Yeast breads and rolls	41.4
9	Yeast breads and rolls	36.4	100% Orange juice	36.8
10	Cakes, cookies, pastries, pies	32.5	Low-calorie fruit juice drinks	36.1

HAZ, height-for-age Z score.

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
