# Peer review of "Nutritional Adequacy and Diet Quality Are Associated with Standardized Height-for-Age among U.S. Children"

_nutrients, 2021, doi:10.3390/nu13051689_

Round 1
Reviewer 1 Report
The Introduction is adequate and describes the background aims of the well.
Lines 79 - 85 The methods section is not described very well, please review this section as I don't fully understand what you have done.
Lines 109 - 115 How have you collected information for the poverty index physical activity, please give more details.
Line 121 when was data on weight collected and how, you have not described this.
Why did you group all the children together, it would be more interesting if you looked at the children by the different age groups reported in table 1.
Results Table 1 : How did you get information on hours of sleep
Results Figure 1 - for energy intake and macro nutrients you did not adjust for BMI or weight . Heavier children will have a greater energy intake. Please give units for energy and macronutrients.
Results Table 3 - give units for nutrients as in table 2. As all the children are grouped together in one age group 2 - 18 how are you comparing to EAR for each nutrient, surely this value will differ by age and gender. You need to explain more how you did this.
Author Response
The authors appreciate the thorough and constructive comments of reviewers, which have helped us to greatly improve this paper. The accompanying manuscript has been reformulated incorporating suggestions from the Nutrients reviewers.
Response to the First Reviewer’s Comments
The Introduction is adequate and describes the background aims of the well.
- Lines 79 - 85 The methods section is not described very well, please review this section as I don't fully understand what you have done.
As advised, more explanation was added in method section. (Page 2, lines 81-91)
- Lines 109 - 115 How have you collected information for the poverty index and physical activity, please give more details.
In NHANES, PIR was defined as ratio of family income to federal poverty level. A PIR of “1” indicates a family income at 100% of the federal poverty level. Eligibility for receiving benefits via the Supplemental Nutrition Assistance Program (SNAP), the premier government food safety net program serving low- and no-income people in the U.S., requires incomes under 130% of the federal poverty level, i.e. a PIR of 1.3. We added more explanation about PIR in the method section. (Page 3, line 122)
We used MET concept provided by 2011 Compendium of Physical Activities. Physical activity was presented as multiplying by the number of days per week by the average minutes of activities on a typical day. We used the MET values provided by NHANES manual, where suggested MET values of walking/bicycling, and moderate recreational activities as 4 and MET value of vigorous recreational activities as 8. We added more explanation about physical activities. (Page 3, lines 127-129)
- Line 121 when was data on weight collected and how, you have not described this.
In NHANES, physical examinations of participants would be conducted in a specially equipped and designed Mobile Examination Center (MEC) on appointment date. This examination includes a physical examination conducted by a physician and laboratory tests, X-rays, and other health measurements and interviews conducted by highly trained medical personnel. We added more explanation in method section. (Page 3, line 104)
- Why did you group all the children together, it would be more interesting if you looked at the children by the different age groups reported in table 1.
We tried to look at odds ratios for the relative risk of intake below the EAR by tertiles of HAZ by three age groups: 2-6, 7-12, and 13-18 years old. However, frequencies of some parts became too small and we could not get reliable results. Thank you for your good comments.
- Results Table 1 : How did you get information on hours of sleep.
In NHANES sleep disorder questionnaires, there is a question of ‘How much sleep do you get (hours)?’. We added more explanation in method section. (Page 3, lines 119-120)
- Results Figure 1 - for energy intake and macro nutrients you did not adjust for BMI or weight. Heavier children will have a greater energy intake. Please give units for energy and macronutrients.
Thank you for your good comments. If we look at diet quality, adjusting for BMI or energy intake may be reasonable. However, we think that energy intake may be an important contributor to children’s growth. In this study, we think that it may be appropriate not to adjust for BMI or weight for intake of energy and macronutrients according to HAZ. We presented units for energy and macronutrients in y-axis of Figure 1.
- Results Table 3 - give units for nutrients as in table 2. As all the children are grouped together in one age group 2 - 18 how are you comparing to EAR for each nutrient, surely this value will differ by age and gender. You need to explain more how you did this.
As advised, we added the units for nutrients in table 3 (Page 6, Table 3). We used EAR for each nutrient by age and gender. As advised, we added more explanation in method section (Page 2, lines 94-95).
Reviewer 2 Report
Nutritional Adequacy and Diet Quality are Associated with 2 Standardized Height-for-Age among U.S. Children
In the paper the association between macro- and micronutrient intake, food consumption and height-for-age Z score (HAZ) of US children based on a data set from NHANES were analyzed and discussed.
The authors referred the literature about the issue, but only including papers, there are in line with the common assumption that nutrition has a direct influence on body height. Recently some papers are published, were other authors do not find any direct association neither in well-nourished populations (Pospisil et al. (2017) doi:10.1127/anthranz/2016/0704) nor in low-and middle-income countries (Mumm (2019) doi: 10.1127/anthranz/2019/0988). At least this should be discussed.
I missed a clear statistic testable hypothesis. What are the predictions of the scientific question?
The results are well presented, but I was wondering about results interpretation.
The sentence (line 169 ff) “In this nationally representative study, US children aged 2-18 years old with higher HAZ tended to have a higher income and BMI percentile” discribes a phenomenon that we have in all populations affected by obesity. Obese children develop faster than slim counterparts do, so the HAZ is always higher, they reach the target height earlier and are not always the tallest in a population as young adults. The reason is not the better availability of micronutrient that they eat more in sum. It is only a matter of tempo.
Otherwise, it is known that additional administration of iron, zinc, calcium, Iodine, Vit A, multiple micronutrient, and protein have only a marginal effect of height increase in populations with a HAZ mean below -2 SDS. That means only in population with more than 50 % stunted children we could observe comparable even if only slightly higher body height after micronutrient supplementation ( Roberts et al. 2017 doi: 10.3945/an.116.013938). US children are not stunted.
The authors mentioned that (again line 169) ” … higher HAZ tended to have a higher income”. Bogin (2021) explains in the 3rd edition of his book “Patterns of Human Growth” that there is strong evidence that height is more affected by social, economic, political, and emotionally (SEPE) factors than by nutrition, especially if there is no extreme hunger situation per se. That is not the case in US.
Maybe the presented results show pseudo relations.
In addition, birthweight is not a genetic factor, so it is not necessary to control the data with birthweight to exclude the genetic factors.
Author Response
The authors appreciate the thorough and constructive comments of reviewers, which have helped us to greatly improve this paper. The accompanying manuscript has been reformulated incorporating suggestions from the Nutrients reviewers.
Response to the Second Reviewer’s Comments
In the paper the association between macro- and micronutrient intake, food consumption and height-for-age Z score (HAZ) of US children based on a data set from NHANES were analyzed and discussed.
- The authors referred the literature about the issue, but only including papers, there are in line with the common assumption that nutrition has a direct influence on body height. Recently some papers are published, were other authors do not find any direct association neither in well-nourished populations (Pospisil et al. (2017) doi:10.1127/anthranz/2016/0704) nor in low-and middle-income countries (Mumm (2019) doi: 10.1127/anthranz/2019/0988). At least this should be discussed.
Pospisil’s pilot study was conducted with only 28 children. Mumm’s study was performed using limited nutrition information, such as only six food groups, no energy and micronutrients data, and nonquantitative measurement of consumed food by household rather than individual level. Therefore, these studies seem somewhat insufficient to refute the common assumption that nutrition has an influence on height. Studies on the effects of nutrients, foods and food groups on skeletal growth are still limited. Therefore, we think that future research is needed to clarify relationship between nutrition and skeletal growth using large sample and longitudinal study design. Thank you for your good comments.
- I missed a clear statistic testable hypothesis. What are the predictions of the scientific question?
The hypothesis of this study is that adequate nutritional intake, diet quality and healthier food choices are positively associated with HAZ.
The results are well presented, but I was wondering about results interpretation.
- The sentence (line 169 ff) “In this nationally representative study, US children aged 2-18 years old with higher HAZ tended to have a higher income and BMI percentile” discribes a phenomenon that we have in all populations affected by obesity. Obese children develop faster than slim counterparts do, so the HAZ is always higher, they reach the target height earlier and are not always the tallest in a population as young adults. The reason is not the better availability of micronutrient that they eat more in sum. It is only a matter of tempo.
Thank you for your good points. As this study is a cross-sectional study, we could not capture long-term trajectory of height. We added more explanation in discussion section (Page 8, lines 187-189).
- Otherwise, it is known that additional administration of iron, zinc, calcium, Iodine, Vit A, multiple micronutrient, and protein have only a marginal effect of height increase in populations with a HAZ mean below -2 SDS. That means only in population with more than 50 % stunted children we could observe comparable even if only slightly higher body height after micronutrient supplementation ( Roberts et al. 2017 doi: 10.3945/an.116.013938). US children are not stunted.
Because growth status may be more affected by the overall nutritional adequacy, not a single nutrient, supplementation of specific micronutrients may not show significant effect on height. Many nutritional supplement studies have been conducted primarily on severely stunted children. However, a study of U.S. population aged 4 years and older showed still considerable prevalence of nutritional inadequacy1. Studies on the effects of nutrients, foods and food groups on skeletal growth among population who has not been severely stunted based on usual dietary intake are not sufficient. Thank you for your good points.
- The authors mentioned that (again line 169) ” … higher HAZ tended to have a higher income”. Bogin (2021) explains in the 3rd edition of his book “Patterns of Human Growth” that there is strong evidence that height is more affected by social, economic, political, and emotionally (SEPE) factors than by nutrition, especially if there is no extreme hunger situation per se. That is not the case in US. Maybe the presented results show pseudo relations.
Thank you for your good comments. Although social, economic, political environment are important determinants to health outcome, but nutritional status may be still important determinant to growth and development for children. Although extreme hunger is likely to be linked to severe malnutrition, well-nourished population may not necessarily achieve nutritional adequacy. A study of U.S. population showed considerable prevalence of inadequacy (%<EAR) in nutrients. 6% of the U.S. children aged 2 to 18 years had a prevalence of inadequacy for vitamin A, 19% for vitamin C, 87% for vitamin D, 81% for vitamin E, 62% for vitamin K, 47% for calcium, 36% for magnesium and 7% for zinc2. A study of U.S. population aged 4 years and older showed similar prevalence of inadequacy1. Nutritional adequacy may play an important role in growth. Studies on the effects of nutrients, foods and food groups on skeletal growth among population who do not have severe malnutrition such as Americans are still limited, suggesting that future research is needed. We added more in introduction section (Page 1, lines 42-45).
- In addition, birthweight is not a genetic factor, so it is not necessary to control the data with birthweight to exclude the genetic factors.
A study showed that genetic factors account for 30-80% of birth weight variance3. Although there are not enough studies on the relationship between birth weight and genetic factors, we think that it would be better to include birth weight as genetic factor in analysis. Thank you for your comment.
References
- Wallace TC, McBurney M, Fulgoni VL, 3rd. Multivitamin/mineral supplement contribution to micronutrient intakes in the United States, 2007-2010. J Am Coll Nutr. 2014;33(2):94-102.
- Fulgoni VL, 3rd, Keast DR, Bailey RL, Dwyer J. Foods, fortificants, and supplements: Where do Americans get their nutrients? J Nutr. 2011;141(10):1847-1854.
- Johnston LB, Clark AJ, Savage MO. Genetic factors contributing to birth weight. Arch Dis Child Fetal Neonatal Ed. 2002;86(1):F2-3.